# Harnessing Next-Generation Sequencing as a Timely and Accurate Second-Tier Screening Test for Newborn Screening of Inborn Errors of Metabolism

**DOI:** 10.3390/ijns10010019

**Published:** 2024-03-05

**Authors:** Toby Chun Hei Chan, Chloe Miu Mak, Matthew Chun Wing Yeung, Eric Chun-Yiu Law, Jana Cheung, Tsz Ki Wong, Vincent Wing-Sang Cheng, Jacky Kwan Ho Lee, Jimmy Chi Lap Wong, Cheuk Wing Fung, Kiran Moti Belaramani, Anne Mei Kwun Kwok, Kwok Yeung Tsang

**Affiliations:** 1Newborn Screening Laboratory, Department of Pathology, Hong Kong Children’s Hospital, Hong Kong SAR, China; cch191@ha.org.hk (T.C.H.C.); ycw186@ha.org.hk (M.C.W.Y.); wtk713@ha.org.hk (T.K.W.); jkb22731@connect.hku.hk (J.K.H.L.); u3579389@connect.hku.hk (J.C.L.W.); kwokyeungt@gmail.com (K.Y.T.); 2Metabolic Medicine Unit, Department of Pediatrics and Adolescent Medicine, Hong Kong Children’s Hospital, Hong Kong SAR, China; fungcw@ha.org.hk (C.W.F.); kiranm@ha.org.hk (K.M.B.); kwokmk@ha.org.hk (A.M.K.K.)

**Keywords:** newborn screening, next-generation sequencing, second-tier test, citrullinemia, dried blood spot

## Abstract

In this study, we evaluated the implementation of a second-tier genetic screening test using an amplicon-based next-generation sequencing (NGS) panel in our laboratory during the period of 1 September 2021 to 31 August 2022 for the newborn screening (NBS) of six conditions for inborn errors of metabolism: citrullinemia type II (MIM #605814), systemic primary carnitine deficiency (MIM #212140), glutaric acidemia type I (MIM #231670), beta-ketothiolase deficiency (#203750), holocarboxylase synthetase deficiency (MIM #253270) and 3-hydroxy-3-methylglutaryl-CoA lyase deficiency (MIM # 246450). The custom-designed NGS panel can detect sequence variants in the relevant genes and also specifically screen for the presence of the hotspot variant IVS16ins3kb of *SLC25A13* by the copy number variant calling algorithm. Genetic second-tier tests were performed for 1.8% of a total of 22,883 NBS samples. The false positive rate for these six conditions after the NGS second-tier test was only 0.017%, and two cases of citrullinemia type II would have been missed as false negatives if only biochemical first-tier testing was performed. The confirmed true positive cases were citrullinemia type II (*n* = 2) and systemic primary carnitine deficiency (*n* = 1). The false positives were later confirmed to be carrier of citrullinemia type II (*n* = 2), carrier of glutaric acidemia type I (*n* = 1) and carrier of systemic primary carnitine deficiency (*n* = 1). There were no false negatives reported. The incorporation of a second-tier genetic screening test by NGS greatly enhanced our program’s performance with 5-working days turn-around time maintained as before. In addition, early genetic information is available at the time of recall to facilitate better clinical management and genetic counseling.

## 1. Introduction

Modern biochemical assays using mass spectrometry and immunoassays in expanded screening for inborn errors of metabolism (IEMs) has revolutionized the field of newborn screening (NBS), enabling early diagnosis and commencement of life-changing therapy in these IEM patients before the development of irreversible metabolic catastrophe [1]. However, the continued expansion of NBS’s scope with increasing numbers of biochemical markers covering an ever-increasing number of IEMs would inevitably give rise to false positive NBS results, which translate to unnecessary medical follow-up, investigation and management as well as negative psychosocial impact on the family [2].

Contemporary approaches to improve false positive rates includes the derivation of secondary screening markers (e.g., C14:1/C12:1 for very long-chain acyl-CoA dehydrogenase deficiency), the development of post-analytical multivariate pattern recognition software by a big data approach (e.g., Collaborative Laboratory Integrated Reports (CLIR), formerly known as Region 4 Stork (R4S) project), and the development of second-tier testing for biochemical markers with high specificity for the condition (e.g., liquid chromatography-mass spectrometry analysis (LC-MS/MS) for 17-hydroxyprogesterone, 21-deoxycortisol and other hormones for congenital adrenal hyperplasia) [3,4,5]. While these approaches have proven success in some conditions, false positive rates remained high for some other conditions, especially for those with low positive predictive value primary markers and no known secondary biochemical markers. Citrin deficiency (CD, MIM #605814, also known as citrullinemia type II), a common IEM locally, exemplifies this issue of a high false positive rate attributed to ASS1 carriers with no effective secondary biochemical marker [6,7]. Maintaining multiple second-tier biochemical tests is also time- and labor-intensive.

The emergence of next-generation sequencing (NGS) has reshaped the diagnostic approach in IEMs in the past decade. Compared to biochemical tests, NGS has a vastly greater potential to screen for a long list of IEM conditions in a single test, and it has been increasingly studied worldwide for its potential implementation in NBS projects, such as Babyseq projects, the UK-NHS-Generation study, the Australian GenSCAN, the Screen4Care EU-IMI project, etc. [8,9,10,11].

In Hong Kong, we have launched a second-tier genetic screening test using NGS on dried blood spot samples for simultaneous screening of six IEMs since September 2011, which dramatically decreased the false positive rate and avoided false negative cases while preserving a timely release of NBS result within 5 working days.

## 2. Materials and Methods

The Newborn Screening Laboratory of Hong Kong Children’s Hospital has been providing newborn screening services for 26 IEMs for all babies born in public hospitals with maternity service. First-tier biochemical tests for amino acids and acylcarnitines were performed by the non-derivatized tandem mass spectrometry method on the dried blood spot (DBS) sample. Samples were analyzed using the MassChrom assay (Chromsystems, Gräfelfing, Germany) from 1 September 2021 to 27 April 2022, while samples from 28 April 2022 onwards were analyzed with the NeoBase 2 assay (Revvity, Turku, Finland). The primary biochemical marker cutoffs were assay-specific to account for the method difference.

The second-tier genetic screening test was launched on 1 September 2021 for simultaneous screening of six IEMs. In brief, NBS cases with primary biochemical markers exceeding the cutoff(s) (Table 1 and Figure 1) would be subjected to second-tier genetic screening testing using a fully validated custom-designed AmpliSeq panel (Illumina, San Diego, CA, USA) performed on the iSeq 100 (Illumina, San Diego, CA, USA) platform with procedures as published by our team [12]. Post-analytically, read alignment, variant filtering and calling were performed using the commercial pipeline in NextGENe software version 2.4.2 (Softgenetics, State College, PA, USA) with alignment to the reference genome (GRCh37/hg19) to generate one mutation report (VCF file) for each included NBS case. Further variant annotation of the VCF files was performed using the Varsome Clinical platform (Saphetor SA, Lausanne, Switzerland) with a gene panel filtering function, so that only the genes relevant to the NBS biochemical marker were retained and annotated. The annotated variants were curated and interpreted by chemical pathologists with years of genetic pathology trainings for variant classification as “Pathogenic”, “Likely Pathogenic” and “Variant with Uncertain Significance” with reference to the latest ACMG guidelines [13]. NBS cases would be recalled as screening positive to a metabolic pediatrician for management if the biochemical and/or genotyping result exceeded the laboratory established cutoff (Table 1). The second-tier genetic screening tests cover six target IEM conditions: citrin deficiency (CD, MIM #605814, also known as citrullinemia type II), carnitine uptake defect (CUD, MIM #212140, also known as systemic primary carnitine deficiency), glutaric acidemia type I (GA1, MIM #231670), beta-ketothiolase deficiency (MIM #203750), holocarboxylase synthetase deficiency (MIM #253270) and 3-hydroxy-3-methylglutaryl-CoA lyase deficiency (MIM #246450). For second-tier genetic testing of samples with elevated citrulline, in order to screen for the hotspot variant of the SLC25A13 gene, IVS16ins3kb, a copy number variation (CNV) calling algorithm for exon 17 was developed and validated with 9 CD patients with IVS16ins3kb confirmed by long-range PCR and 29 control subjects. The presence of one copy of IVS16ins3kb causes allelic dropout of the exon 17 amplicon in our panel design and is detected as a likely positive for IVS16ins3kb if the CNV calling of “exon 17” is decreased to half the expected amount.

Before the availability of second-tier genetic testing, NBS of these six IEMs was performed with first-tier biochemical testing of the respective analytes only, historical cutoffs were listed in Table 1 for reference. 

The NBS algorithm with the second-tier genetic screening test is illustrated in detail in Figure 1 and Table 1. Due to assay change from MassChrom to NeoBase 2, the corresponding first-tier biochemical marker cutoffs listed were adjusted with respect to the method to maintain screening performance (Table 1). For the screening of low free carnitine C0 and high citrulline, biochemical cutoffs were added for immediate recall without proceeding to NGS, i.e., free carnitine C0 < 5.0 μmol/L and/or citrulline > 50 μmol/L, as such extreme values are very suggestive for genuine IEMs. Citrulline > 50 μmol/L is also used as biochemical screening cutoff for two other NBS conditions (citrullinemia type 1 and argininosuccinic acidemia) which were outside the scope of our second-tier genetic test.

For NBS of citrin deficiency, the first-tier citrulline cutoff was lowered from 35 μmol/L to 25 μmol/L (corresponds to 99th percentile) to improve sensitivity. Initially, all cases with ≥ one pathogenic or likely pathogenic variant were recalled. This resulted in two false positives (Cit-3 and Cit-4) within the first month of implementation (1–30 September), owing to the high carrier frequency of citrin deficiency in our population [14]. The genotyping cutoff was refined since 1 October 2021 for borderline high citrulline (≥25 and <35 μmol/L), and only cases with two pathogenic or likely pathogenic variants would be recalled as positive.

All screened positive cases were referred to a metabolic pediatrician for expert management. Full biochemical investigations were performed as appropriate for the suspected conditions, which generally included plasma amino acids, plasma/urine acylcarnitine profiling, urine metabolic profiling, urine reducing substances, etc. Genetic confirmatory testing for the suspected condition was performed with a newly collected EDTA blood sample with informed consent, and parental genetic targeted screenings were performed wherever possible for the determination of variant phasing.

We retrospectively reviewed the electronic medical records of NBS cases with second-tier genetic tests performed from 1 September 2021 to 31 August 2022, and we analytically compared the screening performance and clinical outcome of the present two-tier biochemical genetics approach against biochemical screening alone.

## 3. Results

During the period, 22,883 newborn screening DBS samples underwent first-tier biochemical NBS, and second-tier genetic screening was performed for 1.8% of all samples (*n* = 421, 53% male, 47% female). The most common indications for second-tier tests were elevated citrulline (83.7%, *n* = 355), followed by elevated C5DC-carnitine (13.7%, *n* = 58), decreased free carnitine (1.4%, *n* = 6) and elevated C5OH-carnitine (1.2%, *n* = 5). A total of 424 second-tier genetic screening tests were performed for the 421 NBS cases, with two cases having both elevated citrulline and C5DC-carnitine and one case having elevated citrulline and decreased free carnitine.

Seven cases screened positive after second-tier genetic tests and were referred to a metabolic pediatrician for investigation and management. Three cases were biochemically and genetically confirmed as true positives (CD: 2, CUD:1). Four false positive cases were confirmed as carriers of the target conditions (CD carrier: 2, CUD carrier: 1, GAI carrier: 1) (Figure 1). Overall, the combined recalled rate for the six conditions during the study period was 0.031% (*n* = 7), with a false positive rate of 0.017% (Table 2). The early availability of genotyping results in the true positive cases provided key informative evidence for the urgent initiation of metabolic intervention and proper counseling. The overall turnaround time (TAT) with the implementation of second-tier genetic screening test was within 5 working days (median: 3.5; TAT90: 5). The final newborn screening report is readily available by day 5 to 7 of life of the newborn. The clinical journey of the second-tier positive recalled cases are summarized in Table 2.

## 4. Discussion

Expanded newborn screening for IEM conditions with first-tier biochemical analysis with mass spectrometry has become the standard of care in the majority of NBS programs worldwide. While first-tier testing is rapid and sensitive, false positive NBS cases are not uncommonly encountered, owing to the poor specificity of the biochemical analyte to the target condition. A high false positive rate may cripple the overall efficiency of the NBS program with unnecessary clinic visits, investigations, treatment and adverse psychosocial impact on the family [2]. Second-tier testing of biochemical analytes with higher specificity has been proven successful at drastically reducing the false positive rate in various IEM conditions, for example, LC-MS/MS analysis of 17-hydroxyprogesterone, androstenedione and cortisol for congenital adrenal hyperplasia screening, and LC-MS/MS analysis of methylmalonic acid and methylcitric acid for elevated C3-carnitine [5,15]. With decreasing cost and rapidly evolving genetic technology, there is increasing popularity for exploring genotyping targets as a second-tier option in the NBS field, with proven utility in NBS for cystic fibrosis [16]. Owing to its unique ability to simultaneous sequence for multitudinous genotyping targets in a myriad of genes, NGS has been proposed as panacea in NBS.

To prepare for the incorporation of tiered testing with NGS, we have validated a custom-designed amplicon-based NGS panel of 87 genes with the iSeq platform on DBS samples [12]. The second-tier genetic screening service was launched on 1 September 2021, covering six IEM conditions in the first phase. The streamlined second-tier test can be completed in 2.5 days and is scheduled for batch analysis twice weekly, which helped to uphold the overall NBS turnaround time. Compared to practices applying biochemical cutoffs without a second-tier genetic test, the implementation of tiered testing has greatly improved the false positive rate of the six IEMs from 0.38% to 0.017%. For second-tier positive cases, the genotyping information was made available to the treating metabolic physician within a very short TAT of 3.5 to 5 working days, which has been proven to be very helpful for subsequent counseling. Taking the case of Cit-1 as an example, the very early detection of the homozygosity of the common pathogenic variant c.852_855del in the *SLC25A13* gene would translate to a near-100% probability of citrin deficiency in the NBS subject, allowing the treating physician to confidentially initiate immediate dietary intervention in the early neonatal period without the need to wait for further investigation, potentially saving the patient from catastrophic liver complications [17].

Citrin deficiency is a peculiarly common IEM in Hong Kong, with a carrier rate up to 1 in 40 among southern Chinese people [14]. Early treatment is life-saving and could prevent disastrous complications such as acute liver failure. Notably, newborn screening for citrin deficiency using citrulline as a biomarker has been notoriously challenging, with a high false positive rate owing to the carrier status of citrullinemia type I [6,7]. During the period, 19 cases had citrulline levels above 35 μmol/L, which would have been recalled and resulted in false positive NBS results if second-tier tests were not performed. Moreover, we were also experiencing a high false negative rate of citrin deficiency since the local implementation of expanded NBS (10 cases from the start in 2015 to September 2021). The citrulline levels of these cases ranged from 17 to 34 μmol/L with a median of 26 μmol/L (cutoff: 35 μmol/L). This is because of the fact that DBS samples are typically collected between 24 to 72 h of life in our NBS program, while the citrulline levels of a large proportion of genuine CD patients would not be quite elevated until day 3 to 7 of life and onwards [18]. With second-tier genetic screening in place, we are now able to pick up the two cases of citrin deficiency, Cit-1 (27 μmol/L) and Cit-2 (26 μmol/L), which would have gone undetected with a first-tier test alone. (cutoff: 35 μmol/L) [19]. False negative rates could be further improved if the second-tier genetic test could be applied to a lower citrulline level. There was no false negative case of citrin deficiency during the study period, to our best knowledge. 

The revolutionary development of genetic sequencing in the past decade has changed the landscape of newborn screening. Second-tier genetic testing has been proposed for many conditions in the past, including cystic fibrosis, medium-chain acyl-CoA dehydrogenase deficiency, very-long-chain acyl-CoA dehydrogenase deficiency, long-chain hydroxyacyl-CoA dehydrogenase deficiency/trifunctional protein deficiency, multiple acyl-CoA dehydrogenase deficiency, holocarboxylase synthetase deficiency, biotinidase deficiency, citrin deficiency, etc. [16,20,21]. But most applications were limited to target detection of selected variants or up to single-gene analysis using traditional DNA analysis methods like Sanger sequencing and quantitative polymerase chain reactions. The emergence of NGS technology allows for the simultaneous genetic analysis of multiple genetic targets for a large number of different conditions. In the case of cystic fibrosis, the adoption of tiered approaches with NGS had clearly demonstrated superior screening performance compared to the conventional IRT/DNA screening algorithm, with a quantum jump in positive predictive value from 7.3% to 77.8% in Wisconsin and from 3.7% to 25.2% in New York State [22,23]. The feasibility of first-tier genetic screening with NGS has been examined in several national programs and demonstrated excellent potential for NBS of metabolic disease as well as other Mendelian diseases with no known biochemical screening markers [24,25,26,27]. However, false negative NBS results could happen with first-tier genetic screening alone [28]. 

Taking into account the cost and expertise needed to develop a new NGS assay in an NBS laboratory, some would prefer to modify their existing mass spectrometry method to streamline the program’s screening algorithm. Recently, a highly multiplexed biochemical second-tier test using a hydrophilic interaction liquid chromatography (HILIC) column with mass spectrometry was developed to detect 19 key metabolites in a single universal test for 11 NBS disorders, covering amino-acidopathies, organic acid disorders, fatty acid oxidation disorders, Pompe disease and adrenoleukodystrophy [29]. A metabolomics approach using high-resolution mass spectrometry (HRMS) assisted with machine learning interpretative algorithms represents another potential candidate for multiplex screening. By employing a 121-plex metabolite panel coupled with a trained Random Forest machine learning classifier, Mak J et al. have shown that false positive rates could be reduced by 51–100% for four NBS disorders [30]. Most screening programs are now moving towards a combined or complementary approach with biochemical and NGS screening to optimize their screening performance [14,31,32,33].

## 5. Conclusions

In our one-year experience, the implementation of a second-tier NGS screening test greatly improved our program, with reductions in recall, false positive and false negative rates. Moving forward, we are continuing to evaluate including more conditions for NGS second-tier tests, which can hopefully benefit future newborns. There are, nevertheless, challenges with NGS second-tier panel tests, such as the detection of variants of uncertain pathogenicity, limited information on the phasing of detected variants, predominant coverage of only exonic regions and difficulties associated with the detection of complex structural variants. Ongoing studies on whole-genome sequencing approaches and/or third-generation long-read sequencing in the NBS field would definitely shed light on the best newborn screening strategy in the future. But until then, our center has found that a two-tiered approach with NGS is the current best-balanced solution for this multifaceted situation.

## Figures and Tables

**Figure 1 IJNS-10-00019-f001:**
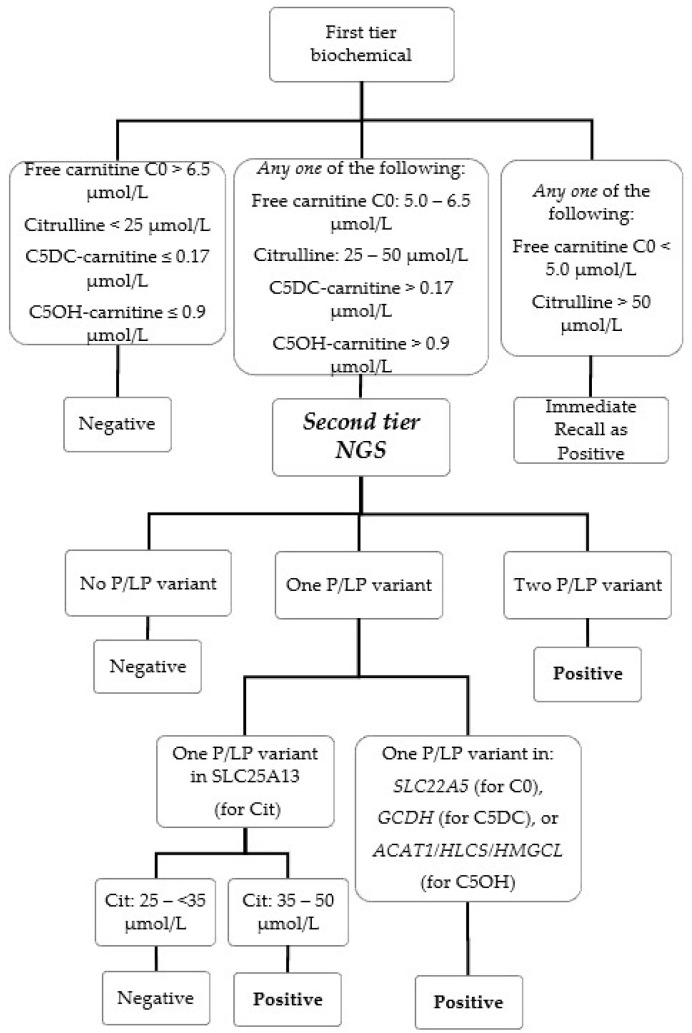
NBS algorithm with second-tier NGS.

**Table 1 IJNS-10-00019-t001:** First-tier biochemical marker, second-tier genotyping markers and target conditions.

First-Tier Biochemical		Second-Tier Genetic		Target Condition	Historical Biochemical Cutoff for Recall without 2nd Tier (μmol/L)
Biochemical Marker	Cutoff (μmol/L)	Genotyping Marker	Cutoff		
Free carnitine-C0	5.0–6.5 ^n/c^ *	Sequence variant in *SLC22A5* gene	≥one P/LP variant	Carnitine uptake defect (CUD, MIM #212140, also known as systemic primary carnitine deficiency)	≤6.2 ^c^
Citrulline	25–50 ^n/c^ ^	Sequence variant and target screening for IVS16ins3kb in *SLC25A13* gene ^$^	1. ≥one P/LP variant and citrulline ≥ 35 μmol/L, or 2. ≥two P/LP variant and citrulline ≥25 and <35 μmol/L ^@^	Citrin deficiency (CD, MIM #605814, also known as citrullinemia type II)	≥35 ^c^
C5DC-carnitine	>0.17 ^n^ >0.35 ^c^	Sequence variant in *GCDH* gene	≥one P/LP variant	Glutaric acidemia type I (MIM #231670)	≥0.35 ^c^
C5OH-carnitine	>0.9 ^n^ >0.7 ^c^	Sequence variant in *ACAT1*, *HLCS* and *HMGCL* gene	≥one P/LP variant	Beta-ketothiolase deficiency (#203750) Holocarboxylase synthetase deficiency (MIM #253270) 3-Hydroxy-3-methylglutaryl-CoA lyase deficiency (MIM #246450).	≥0.78 ^c^

Notes: “n”: NeoBase2 cutoff. “c”: MassChrom cutoff. “n/c”: same cutoff for NeoBase 2 and MassChrom. “P”: pathogenic. “LP”: likely pathogenic. “*”: NBS cases with free carnitine C0 <5.0 μmol/L are recalled immediately as positive and excluded from second-tier genetic screening test. “^”: NBS cases with citrulline >50 μmol/L are recalled immediately as positive and excluded from second-tier genetic screening test. “$”: Target screening for IVS16ins3kb in *SLC25A13* gene was implemented since May 2023; all cases positive for citrulline elevation before May 2023 were retrospectively reviewed. “@”: During 1–30 September 2021, cases with citrulline ≥25 and <35 μmol/L and one P or LP variant would be recalled as screening positive; the second-tier genetic screening cutoff was later refined to “≥two P or LP variant and citrulline ≥25 and <35 μmol/L” to prevent excessive recalls of CD carrier.

**Table 2 IJNS-10-00019-t002:** Clinical summary of NBS recalled cases.

Case Number	Newborn Screening Result		Confirmatory Genetic Test Result	Clinical Summary
	First-Tier Biochemical	Second-Tier Genetic		
Cit-1 (TP)	Citrulline: 27 μmol/L	Homozygous for the pathogenic variant: c.852_855del in *SLC25A13* gene	Genetically confirmed as CD, both parents were heterozygote carrier.	21 months old during latest clinic visit with satisfactory growth and development, on lactose-free formula, uneventful clinical course.
Cit-2 (TP)	Citrulline: 26 μmol/L	Heterozygous for the pathogenic variant, c.1311+1G>A, and Target screening positive for heterozygous IVS16ins3kb variant, in *SLC25A13* gene	Genetically confirmed as CD, both parents were heterozygote carrier.	19 months old during latest visit, now with satisfactory growth and development, on lactose-free MCT-enriched formula. Initially screened negative with sequence analysis alone. Presented with conjugated hyperbilirubinemia and failure to thrive at 2 months while on breastfeeding, with no definitive diagnosis despite extensive investigation performed including plasma amino acids and liver biopsy. Jaundice resolved spontaneously at 3 months. Recalled and eventually diagnosed as citrin deficiency at 13 months of age after implementation of target screening for IVS16ins3kb variant. Catch-up growth with appropriate dietary management.
Cit-3 (FP)	Citrulline: 26 μmol/L	Heterozygous for the pathogenic variant, c.615+5G>A	Genetically confirmed as CD carrier.	Case closed at 7 months old during latest clinic visit after extensive biochemical and genetic investigations, normal growth and development.
Cit-4 (FP)	Citrulline: 26 μmol/L	Heterozygous for the pathogenic variant, c.615 + 5G>A	Genetically confirmed as CD carrier.	Case closed at 7 months old during latest clinic visit after extensive biochemical and genetic investigations, normal growth and development.
C0-1 (TP)	Free carnitine C0: 6.9 μmol/L Citrulline: 26 μmol/L	Heterozygous for the pathogenic variant, c.51C>G, and Heterozygous for the pathogenic variant, c.1400C>G in the *SLC22A5* gene No P/LP variants in the *SLC25A13* gene	Genetically confirmed as CUD, both parents were heterozygote carrier.	12 months old during latest clinic visit with satisfactory growth and development, on levocarnitine, uneventful clinical course
C0-2 (FP)	Free carnitine C0: 6.3 μmol/L	Heterozygous for the pathogenic variant, c.1400C>G in the *SLC22A5* gene	Genetically confirmed as CUD carrier	Case closed at 3 months old during latest clinic visit after full biochemical and genetic investigations, normal growth and development.
C5DC-1 (FP)	C5DC-carnitine: 0.18 μmol/L	Heterozygous for the pathogenic variant, c.1156C>T in the *GCDH* gene	Genetically confirmed as GA1 carrier	Case closed at 7 months old during latest clinic visit after full biochemical and genetic investigations, normal growth and development.

## Data Availability

The data presented in this study are available on request from the corresponding author. The data are not publicly available due to patient privacy issues.

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
