# Peer review of "Harnessing Next-Generation Sequencing as a Timely and Accurate Second-Tier Screening Test for Newborn Screening of Inborn Errors of Metabolism"

_2409-515X, 2024, doi:10.3390/ijns10010019_

Round 1

Reviewer 1 Report

Comments and Suggestions for Authors

In this paper, the authors report their first-year experience and outcome of implementing genetic testing as a second-tier test in newborn screening (NBS) for six inborn errors of metabolism using next generation sequencing technology.  It is not a novel concept that multiple-tier testing in NBS can be helpful in both testing sensitivity and septicity, but this paper still holds its value because the application of next generation sequencing in routine NBS practice is in its infancy, especially when targeting multiple disorders and multiple specimens simultaneously.  I would encourage the authors to change the statement of “But these were limited to target detection of selected variants or up to single gene analysis using traditional DNA analysis like Sanger sequencing and quantitative polymerase chain reaction,”  because there are reported experiences in routine NBS for cystic fibrosis using next generation sequencing technology.

Reviewer 2 Report

Comments and Suggestions for Authors

The manuscript by Chun Hei, et al., presents the potential utility of a second-tier molecular assay for 6 IEMs commonly screened for via Newborn Screening. While, certainly, the potential value of molecular information in NBS is a contemporary topic, the manuscript could be improved with the following edits:

1) There should be more discussion of the role of 2nd tier biochemical assays and how these have greatly improved PPV of screening results. Including new attempts to multi-plex second-tier assays to mitigate issues with maintaining multiple second-tier assays (https://pubmed.ncbi.nlm.nih.gov/36724346/).

2) Many programs also utilize secondary analytes or ratios to bring down FPRs; it is not clear what the PPV would have been using these and how this might compare to the use of NGS.

3) Table 1 could use some formatting assistance as one column is larger and difficult to read

4) Unclear what "ve" stands for in Figure 1

5) Please indicate the time from birth to results rather than working days. Given the very time critical nature of some of these diseases (like Cit); days are paramount and it is helpful to understand at what day of life these results would be available to clinicians.

6) The assertion in the Conclusion regarding cost justifications should be supported by some data or a reference. As it is not clear what the cost for the NGS second-tier assay is; it is hard to determine if this sentence is appropriate or not.

Comments on the Quality of English Language

There are minor edits to English that are needed to improve readability of the manuscript, but overall, is well-written.
